# Characterization of Hard Coatings Using Acoustic Emission

**DOI:** 10.3390/ma18163777

**Published:** 2025-08-12

**Authors:** Ivana Sára Škrobáková, Peter Gogola, Marián Palcut, Ľubomír Čaplovič

**Affiliations:** Institute of Materials Science, Faculty of Materials Science and Technology in Trnava, Slovak University of Technology in Bratislava, Ulica Jána Bottu 25, 917 24 Trnava, Slovakia; ivana.skrobakova@stuba.sk (I.S.Š.); peter.gogola@stuba.sk (P.G.); caplovic@chello.sk (Ľ.Č.)

**Keywords:** acoustic emission, indentation testing, adhesion, PVD coating, CrN, SEM

## Abstract

Acoustic emission (AE) testing is a non-destructive method used in various applications. In our work we demonstrate its capabilities and potential in studying the functional properties of physical vapor deposited (PVD) coatings. The goal was to classify the coating damage during indentation testing more objectively by quantifying specific imprint features. The AE response was systematically recorded in nine sample conditions and 27 individual imprints, allowing us to identify correlations between the numerical values derived from the SEM observations and the characteristics of the AE signal. An increase in the delaminated coating area was found to correspond to an exponential increase in the AE signal energy. These findings suggest that AE analysis could reduce the reliance on SEM-based evaluation and help accelerate systematic research in the field of PVD coatings. The advantages of AE testing are discussed and conclusions for practical applications are provided.

## 1. Introduction

Modern coating research efforts require gradual improvements in the development of new coating technologies and optimization of existing solutions. Physical vapor deposited (PVD) coatings possess exceptional hardness and wear resistance. Coating adhesion is a critical property; however, it lacks a single fully standardized and easily repeatable quantification method. Instead, various descriptive techniques are used, often based on how the coating/substrate system responds to the applied load. Scratch and indentation tests are most commonly used to assess the adhesion through failure modes such as cracking or delamination [1,2,3,4].

During indentation testing, the indenter penetrates the sample surface. According to the applied load and penetration depth, macro- (load over 2 N), micro- (load 10 mN–2 N, depth ~1 µm) and nano-indentation (load less than 10 mN, depth within the nanometer range) can be distinguished [5]. The behavior of a coating during a macro-indentation test is one of the key parameters to be evaluated. The Mercedes (Daimler–Benz) adhesion test with an applied load of up to 1500 N is considered a macro-indentation test. The indentation causes damage to the coating by forming an imprint (also denoted as an indent) around which cracks and de-lamination (also described as flaking or spalling) can form. Evaluation of these imprints enables the assessment of adhesion according to the standards [6,7]. Cracks and coating delamination are key indicators of the coating’s adhesion. The most widely referenced standard is VDI 3198 [8]. It classifies the adhesion strength (HF = Haftfestigkeit) between HF1 (minimal damage; the best adhesion of the coating with only radial cracking) and HF6 (extensive coating delamination; very poor adhesion) [6,7,8].

The types of damage given above are typically evaluated using microscopy. The use of either light or scanning electron microscopy (SEM) allows a quick assessment of the coating adhesion behavior according to the VDI HF classification. However, microscopy as an analytical technique has a limited field of view, making it difficult to assess large areas or structures accurately. The operation of high-resolution, high-vacuum instruments, such as SEM, can also be expensive and destructive to some materials. Acoustic emission (AE) represents an alternative approach. The AE testing is a valuable tool for real-time, non-destructive monitoring of materials and structures, particularly for early defect detection and continuous assessment [9]. AE can capture the onset and evolution of damage without interrupting the indentation process. This approach enhances the value of Rockwell indentation in industrial applications by providing quantitative data on the progression of coating damage, complementing traditional visual inspection methods.

Cracks can be formed on multiple sites in the vicinity of an imprint and consequently propagate in different directions. These cracks can be classified as radial cracks, circumferential cracks, cracks along intercolumnar boundaries (shear cracks), trans-columnar cracks (lateral, edge and bending, lateral cracks) and cracks from the interface [10]. If the maximum tensile strength of the coating is exceeded during indentation, the cracks appear on the coating surface at the edge of the imprint and then may propagate outward. These cracks are visible as radial cracks. The cracks are most often observed around the imprint, where the so-called pile-up of the substrate material causes tensile stresses in the coating [11]. An AE event is triggered when the voltage amplitude of a detected signal exceeds a predefined threshold. Each event, referred to as a “hit”, corresponds to one or more instances of cracking or other structural instabilities. The energy released during these events is quantified as “hit energy”, which accounts for both the intensity and duration of the signal.

Pile-up is the plastic deformation zone where the material around an imprint (in-dent) is built up above the original surface level. This region of an imprint of an indentation test is more closely characterized in the case of nano-indentation and scratch test experiments; however, it is also commonly observed to form after macro-indentation measurements [11,12]. Circumferential cracks usually appear on the free surface outside the indenter contact area. They are primarily parallel to the edge of the imprint, and their propagation direction often differs from the original direction of the indenter axis. Most of these cracks are caused by the strain incompatibility between the coating and the substrate—their mismatch is in their stress field rather than their microstructure [13].

Further failure modes may occur during the indentation including delamination (the loss of contact between a coating and its substrate), buckling (the deflection of a delaminated coating away from the substrate due to stress), and spallation (exposing the underlying substrate). All these phenomena can be considered localized rearrangements within the microstructure of the material. These local material changes generate transient elastic stress waves, which are known as acoustic emission (AE) events. During instrumented indentation, AE events cause the release of elastic energy, which propagates as stress waves and is detected by piezoelectric sensors. The resulting electrical signals are referred to as AE signals and are recorded by specialized equipment [14,15].

Based on this understanding, it was assumed that all structural instabilities forming gradually would generate less significant AE signals (hits), e.g., radial cracks. For example, so-called radial cracks would fall into this category. On the contrary, suddenly occurring defects, such as coating delamination (particularly where the coating separates from the substrate), would produce a strong AE response. Similarly, “planar” cracks, which propagate nearly parallel to the surface and may result in partial coating chipping, are expected to generate comparable AE signals. Therefore, further evaluation of these damage characteristics was found to be important in supporting and refining the interpretation of AE measurements [4,14].

In this work, a set of cemented carbide substrates with varying surface roughness was utilized to consider a range of conditions. A CrN type coating was chosen, as it is a rather well-known coating system. Hence, the results could be easily compared with available literature data. The novelty of our work lies in the combination of quantitative SEM analysis with acoustic emission (AE) monitoring to correlate the delamination behavior during the macro-indentation of PVD CrN coatings. Although AE has been used in micro-scratch tests previously [14,15], its application in macro-indentation under high loads (up to 1500 N) remains unexplored. Multiple factors, such as mechanical interlocking [16], influence the adhesion between the coating and the substrate. Controlling surface roughness can significantly influence coating adhesion. In many cases, increased substrate roughness has been reported to improve adhesion [17,18,19]. However, this effect strongly depends on the specific coating system, particularly the combination of substrate materials and PVD coatings. Studies have also examined the adhesion of the coating across a range of surface roughness levels. An arithmetical mean roughness (Ra) value of approximately 0.1 µm has been identified as optimal in several instances, while lower and especially higher roughness values were associated with reduced adhesion [20,21]. Specifically, for tungsten carbide (WC) substrates, laser ablation was found to improve not only the mechanical anchoring (interlocking) of the coating but also the crystallographic orientation of the substrate surface grains, thus enhancing the potential for coating growth with improved adhesion [22,23,24]. SiC matrix composites were also investigated using AE in a fatigue regime at different frequencies [25]. Identifying when and where AE events occurred, coupled with waveform analysis, led to source and failure progression identification. The AE was able to distinguish the different types of progression and damage via location, energy, frequency, and direction identification of the first extensional peak of the waveform [25].

As the present study aimed to explore the feasibility of using AE measurements to support indentation-based adhesion testing, substrates with surface roughness values both within and deliberately outside the ranges suggested by the literature were prepared. This approach was chosen to ensure a broad spectrum of adhesion performance in all tested samples. Samples with well-adhering coatings and some with sub-optimal coating adhesion were expected. Following this approach, it was possible to evaluate the relevance of AE signals to distinguish between varying degrees of coating adhesion more reliably.

## 2. Materials and Methods

The Boehlerit HB30F (Boehlerit GmbH & Co. KG, Kapfenberg, Austria) tungsten carbide substrate containing 10 wt.% of Co was selected for the experiments. The measured hardness was 1940 HV1, which is consistent with the literature [26]. Before coating deposition, the samples were treated by laser ablation. A SPI Laser, a G4 Pulsed Fiber Laser system (Trumpf, Ditzingen, Germany), was utilized. For each sample set, different laser ablation parameters were applied, as summarized in Table 1. Parameters were chosen to produce surfaces with a range of roughness values [21,27]. In total, nine conditions were prepared. For each condition, three individual surface roughness measurements were carried out, each covering an area of approx. 300 × 300 µm. These measurements were carried out before and after coating deposition. Cylindrical samples were Ø12 mm in diameter and 4 mm in height, with an 8 × 8 mm square area modified by laser ablation.

Coating deposition was carried out on a PLATIT π80 + DLC deposition unit (Platit, Selzach, Switzerland) using a Lateral Rotating Cathode (LARC^®^) system. Process parameters are detailed in Table 2. A single 99.99 wt.% Cr cathode was used, with the deposition conducted in a flowing pure nitrogen atmosphere.

Surface roughness was measured using a ZEISS LSM700 laser scanning confocal microscope (LSCM, Carl Zeiss AG, Oberkochen, Germany). A 405 nm laser light source was utilized, in combination with an Epiplan-Apochromat 20×/0.50 objective. The same setup was later used to investigate the imprint topography.

The adhesion of CrN coatings was evaluated using the Daimler–Benz test on a UMZ-3K Rockwell hardness tester (Micro-Epsilon, Bechyně, Czech Republic), using a conical diamond indenter (120°). A series of Rockwell indentation tests were performed on each sample at maximum loads of 50 N, 500 N, and 1500 N. Multiple imprints were created and examined at each load. The imprint classification was done on the VDI 3198 standard. Although DIN EN ISO 26443 is a more recent standard, the slightly more granular classification system defined in VDI 3198 remains prevalent in the literature [8]. This is exemplified by multiple studies executing similar tests on CrN coatings applying loads up to 1500 N [4,8,28,29]. Therefore, the VDI 3198 standard was used.

The highest-load tests (1500 N) were augmented by acoustic emission (AE) measurements. The AE signals were recorded via a DAKEL-ZEDO device (ZD Rpety, Rpety-Hořovice, Czech Republic). The sample was placed on a custom-made AE detector (holder type) incorporating a piezoelectric sensor. The experimental setup is shown in Figure 1. A tiny amount of silicone grease was used to provide better transfer of the acoustic signals with the smallest possible loss. A 5.0 MHz sampling frequency was selected as the most significant frequency range for these AE measurements [15]. For each sample, the 1500 N load imprints and accompanying AE measurements were repeated three times. Each record was evaluated separately and the data for each condition were given as an average of these data.

SEM imaging was performed on a JEOL JSM 7600F scanning electron microscope (SEM, Jeol Ltd., Tokyo, Japan) with a Schottky field emission electron source operating at 20 kV. Samples were placed at a working distance of 15 mm and documented using both an in-axis and an off-axis backscattered electron detector (BSE). Hence, two sets of BSE images were recorded. Images from the off-axis BSE detector were used to more reliably recognize the imprint edges and the extent of the delamination damage. Images of the in-axis detector enabled more reliable crack detection. Images for the image analysis were recorded at a resolution of 5120 × 3840 pixels (19.66 Mpixels) at a magnification suitable to observe the entire imprint, including all damages (cracks and delamination), in a single view while ensuring reliable recognition of features down to the micrometer scale.

The length of the radial cracks was analyzed with the help of ImageJ FIJI software [30] by measuring the length of each crack between the edge of the imprint and the crack tip. Delaminated areas were manually masked using ImageJ Version 1.54p. The image masks were drawn wherever the substrate was exposed or the coating was at least partially chipped. Furthermore, damages inside and outside the indentation imprint were distinguished. This resulted in a set of B/W masks that were analyzed, and corresponding areas in µm^2^ were obtained.

## 3. Results

The sample roughness was estimated both before and after PVD coating deposition. The acquired data are summarized in Figure 2. Standard deviations are indicated as error bars. The arithmetical mean surface roughness values (Sa) ranged from 0.1 to almost 3 µm, while the coated samples showed lower roughness values from 0.1 to 0.5 µm.

After indentation testing, all imprints were investigated with respect to their geometry, including the extent of the coating damage by means of SEM and LSCM. Some examples of the SEM images are shown in Figure 3. This set of images compares the samples with varying laser powers and pulsing frequencies. The morphology shown in Figure 3 possesses a certain degree of randomness. It can be influenced by selected regions and the surrounding environment. Therefore, the length of the radial cracks was analyzed with the help of ImageJ by measuring the length of each crack between the edge of the imprint edge and the crack tip. Delaminated areas were manually masked. The image masks were drawn wherever the substrate was exposed, or the coating was at least partially chipped. Furthermore, damages inside and outside the indentation imprint were distinguished. This resulted in a set of B/W masks that were analyzed, and corresponding areas in µm^2^ were obtained.

Examples of processed images with their imprint features highlighted are shown in Figure 4. The extent of the radial cracks was easier to distinguish in the images of the in-axis BSE detector (exemplified in Figure 4a), while the off-axis BSE detector (Figure 4b) was much better suited to identify the delamination damage. Combining both detector types enabled identification of the delamination damage where the coating was removed either completely or partially.

With the help of LSCM, the depth profile of the imprints was obtained, and the formation of a pile-up region was confirmed for all imprints investigated. An example of this measurement is given in Figure 5 with the pile-up region highlighted (Figure 5a) and further confirmed in the corresponding height-profile chart (Figure 5b).

The imprint diameters were also measured and an average for all measurements at the individual load levels was plotted in relation to the duration of the test. The results are presented in Figure 6. Standard deviations are indicated as error bars. This approach allowed us to assess the edge of the speed at which the imprint was propagating. At the same time, we could identify that the average radial crack length closely followed the extent of the pile-up region that was obviously gradually being pushed outward by the propagating imprint edge. These data together allowed us to assess the time period responsible for causing most of the coating damage observed outside the imprint edges that were not obscured by further propagation of the imprint.

All 1500 N indentation tests were accompanied by AE measurements. The energy associated with each hit could be summed for the duration of the indentation test, resulting in a cumulative hit energy chart. This relationship is illustrated in Figure 7.

## 4. Discussion

The variations in the laser parameters led to a range of substrate surface roughness values, which in turn affected the coating adhesion. The laser ablation parameters clearly influenced the substrate roughness, with the laser power and repetition rate having the most significant effect, while the laser spot speed showed an almost negligible influence. According to our results, the roughness of the samples decreased after deposition (Figure 2). We conclude that the coating was initially deposited in the valleys of the substrate surface, effectively filling them, and resulting in a smoother final coating surface. This aligns with observations reported in previous publications [21,31,32].

The coated samples were subjected to macro-indentation testing. A series of SEM BSE images were acquired for all imprints, allowing classification according to the VDI 3198 standard and further detailed image analysis (Figure 3). Radial cracks and areas of delamination of varying degrees were observed in the vicinity of almost all imprints. Most of this coating damage (both radial cracks and delamination) occurred in the region commonly known as pile-up [10]. The presence of such a region is exemplified on an LSCM scan in Figure 5. While the VDI 3198 standard typically requires optical microscopy (OM) images to classify coating damage outside the edge of the imprint, the SEM images used in this study provided a significantly higher depth of focus and higher resolution, allowing for a more detailed analysis of each imprint. The delaminated area, only outside the indent, was evaluated to align with VDI 3198 requirements. This is illustrated in Figure 4b by summing the teal and mustard-colored regions.

A comparison between the area of delamination outside of an imprint and the corresponding HF classification is shown in Figure 8a. Due to the limitation of the standard to six discrete classes, the VDI 3198 HF scale lacks sufficient resolution to clearly differentiate samples with more subtle differences. On the contrary, quantifying delaminated areas through SEM imaging, though more time consuming, allows for finer distinctions.

As noted in the introduction, AE offers an alternative evaluation method. It is less time-consuming than detailed SEM analysis and provides a higher resolution than the VDI HF scale. The main drawback of AE is the need for a more complex experimental setup and parameter optimization, which is particularly worthwhile when testing of larger sample sets is required. AE measurements require careful tuning of hit detection parameters, including gain settings, threshold levels, and time-related parameters such as minimum and maximum hit duration, all of which must remain consistent across measurements to ensure reliable comparisons between samples and repeats. The substrate material, coating type, thickness, and even sample shape can require a renewed fine-tuning of these parameters. As noted in a recent review, AE captures elastic waves released from localized failure sources within composite materials during mechanical loading [9]. As such, it allows for early identification of failure modes [9,25]. Furthermore, by using AE it is also possible to monitor the damage progression in real time. However, AE requires continuous monitoring, and the interpretation can be affected by background noise. Therefore, the combined use of AE and SEM to assess crack initiation and progression is beneficial. The present study demonstrates that the need for visual inspection via OM and/or SEM can be reduced. As such, the indentation testing of materials can be significantly accelerated.

In this study, the AE signals were recorded along with all 1500 N indentations, and as a result, a correlation between the SEM image analysis and the AE hit energy data was found (Figure 7). It must be noted, however, that while the AE data are recorded continuously during the test, the SEM evaluation occurs post-test. As such, later plastic deformation under the indenter may obscure the coating damage that occurred in earlier stages. To reduce this uncertainty, additional values were calculated from the SEM images, including the average length of the radial cracks and the approximate imprint expansion rate during the test. These metrics are closely related to the pile-up phenomenon described earlier. Radial cracks consistently followed the pile-up region, with their average length increasing only slightly from 37 to 43 µm (Figure 6), while the imprint diameter expanded from 180 µm to 330 µm (Figure 6), following the imprint data of 500 N and 1500 N. Based on these data, it was concluded that all damage around the imprint occurred during the final ~63 s of the test. Any coating damage that occurred during this period was never obscured by the expanding imprint itself. Thus, the AE cumulative hit energy from the last approx. 63 s should be most reliably linked to the coating damage around an imprint after the complete 1500 N test.

AE data were plotted against the delaminated area outside the imprint. The results are presented in Figure 8b. A strong, apparently exponential, correlation was found. As the delamination increased, a significant increase in the intensity of the AE signal was observed. To most reliably explain the recorded AE signals, the combined area of both full and partial delamination (mustard and teal regions in Figure 4b) had to be considered, as both types of damage generated significant AE responses. This suggests that both full and partial delamination contribute substantially to the AE signal, and that their combined area should be considered to accurately interpret the recorded data.

The relationship between the cumulative hit energy of acoustic emission and delaminated coating area measured outside the indentation imprint is exponential (Figure 8b). It can be expressed by Equation (1).(1)$$E_{CHE}=E_{0}\cdot e^{k\cdot A_{d}}$$

In Equation (1) E_CHE_ represents the cumulative acoustic emission hit energy; E_0_ is the baseline AE energy level controlled by AE hit-detection threshold levels; A_d_ is the delaminated coating area measured outside the indentation imprint and k is a fitting constant describing the sensitivity of the AE energy to changes in the delaminated area. For the currently investigated data set, the constants E_0_ and k were found to be E_0_ = 0.33 μV^2^ s and k = 0.0003 μm^−2^. The observed exponential relationship (Equation (1)) may be attributed to the characteristics of delamination damage, where the most intense AE signals are generated during widespread delamination or low-angle spalling. In such cases, the AE signals are not confined to narrow cracks but are emitted over a broader surface area.

By comparing Figure 8a,b, one can observe that samples with a vastly different delaminated area can be classified very similarly via VDI 3198, as shown for the samples 2B, 1B, 1C, and 2A, while the same samples can be better distinguished when relying on AE data. In general, AE appears to be more useful in nano-indentation applications, where fewer AE-generating events occur. In contrast, for macro-indentation measurements, focusing on individual AE hits can be less informative due to the complex overlapping signals. Therefore, it seems more appropriate to rely on cumulative hit counts or cumulative hit energy when analyzing AE data from macro-indentation tests.

Faisal et al. summarized data from multiple sources where nano-indentation measurements were successfully combined with AE [14]. However, very few publications addressed the application of AE in combination with Rockwell macro-indentation for coating adhesion testing. For example, Drobný et al. [4] reported a successful proof of concept on a small set of samples, demonstrating that AE measurements can provide valuable information about the extent of coating damage during such tests. Similarly, Belmonte et al. [18] successfully identified spalling during macro-indentation of DLC coatings on Si_3_N_4_ substrates. However, none of these publications attempted to link the extent of damage to the intensity of the AE signal.

AE measurements are commonly applied in the scratch testing of coatings [33,34]. For example, Tomastik et al. studied the scratch testing of SiC-based coatings and found that a good correlation could be achieved because the entire scratch track is visible after the test [34]. However, they also noted that additional coating failures can occur even after the indenter has moved on, complicating the precise correlation with the AE data. Therefore, the AE information is usually validated by off-line damage inspection. Our observations (Figure 8) indicate that the ongoing challenges of exactly linking AE signals with visually observable damage may be reduced. The fine tuning of the AE measurement parameters can enable one to increase the number of experiments without significantly increasing the need for detailed SEM analysis, or even OM, while improving the objectivity of the acquired results.

Overall, the coating adhesion was good across most samples. Where delamination from the substrate occurred, it was often accompanied by low-angle spalling of the coating, suggesting that the adhesion strength may be comparable to the tensile strength. In general, smoother substrate surfaces led to better adhesion performance. It must be also acknowledged that the established relations were empirical in view of the input conditions that influenced the AE measurement data. The conclusions are thus linked to the specified substrate/coating combination (WC/CrN) only.

## 5. Conclusions

The potential of acoustic emission (AE) measurements to complement coating adhesion testing via the Daimler–Benz Rockwell macro-indentation method was investigated. It was found that the delaminated area around the imprint is well-correlated with the cumulative AE hit energy. Both full delamination and low-angle spalling must be considered to obtain the most reliable correlation. This observation is also in accordance with the classification system proposed by the VDI 3198 standard [8].

Our findings are consistent with previous studies that have demonstrated a relationship between coating damage and acoustic emission (AE) signals in PVD coatings [34,35,36]. However, in contrast to earlier work that focused primarily on lower loads, this study extended the use of AE monitoring to macro-indentation at higher loads, incorporating a detailed correlation with SEM-based delamination quantification. Additionally, the sample size was substantially increased. This broader approach has enabled a more systematic investigation of the cracking phenomenon.

It can be concluded that incorporating AE measurements alongside macro-indentation testing can enhance the evaluation process, allowing for multiple repeats without significant additional effort. Furthermore, the AE provides a more objective and quantifiable approach compared to the traditionally used, but more subjective, VDI 3198 classification.

This study also demonstrates for the first time that a clear correlation exists between the cumulative AE energy and coating delamination area during high-load Rockwell macro-indentation. This result is promising and demonstrates that an objective and efficient alternative to existing adhesion assessment methods can be found. The future development of this user-friendly framework for processing the AE data may facilitate scientific measurements and accelerate standard testing. The follow-up work is planned. It will include the AE testing of several coated material systems including high-speed steel (HSS) substrates and alternative coatings including TiAlN. Additionally, a more detailed evaluation of the recorded AE signals will be attempted.

## Figures and Tables

**Figure 1 materials-18-03777-f001:**
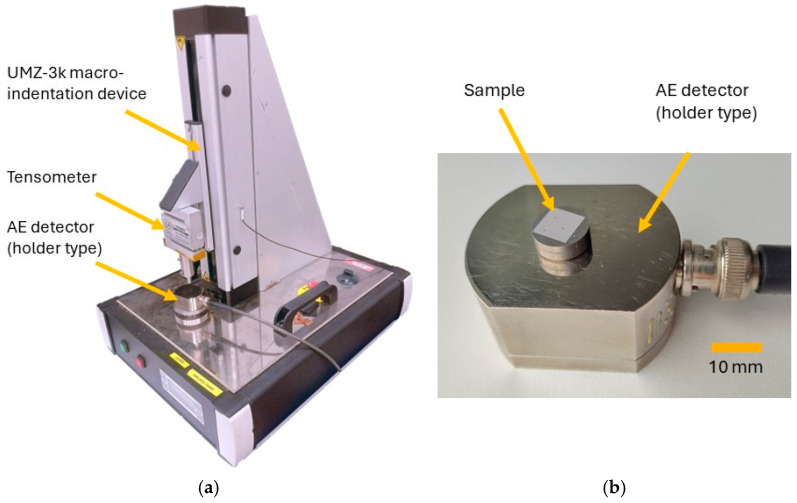
Experimental setup utilized for indentation testing: (**a**) experimental setup for indentation testing; (**b**) detail of a sample positioned on the AE detector.

**Figure 2 materials-18-03777-f002:**
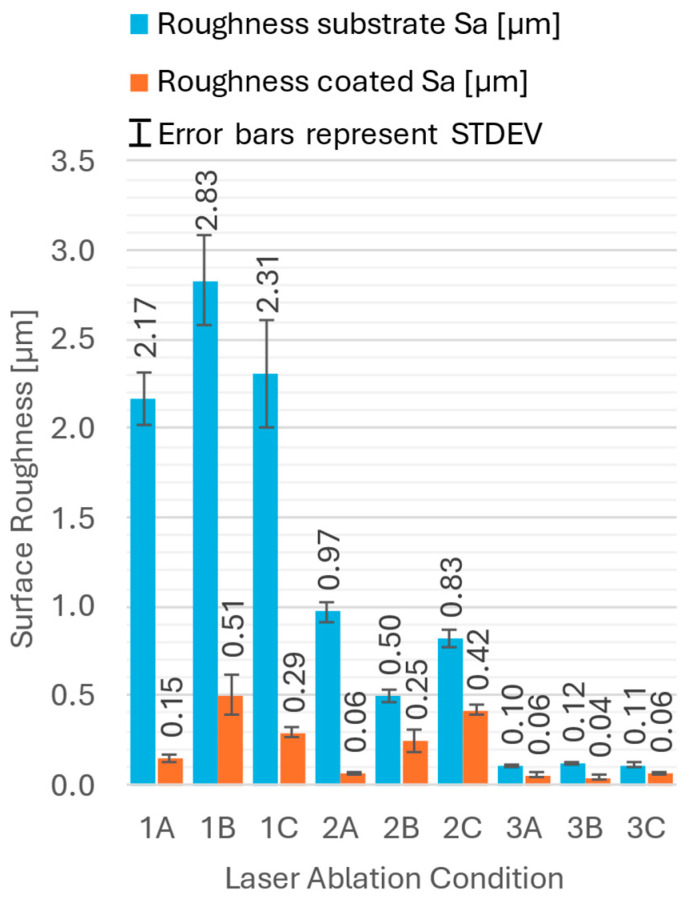
Substrate and coating surface roughness values.

**Figure 3 materials-18-03777-f003:**
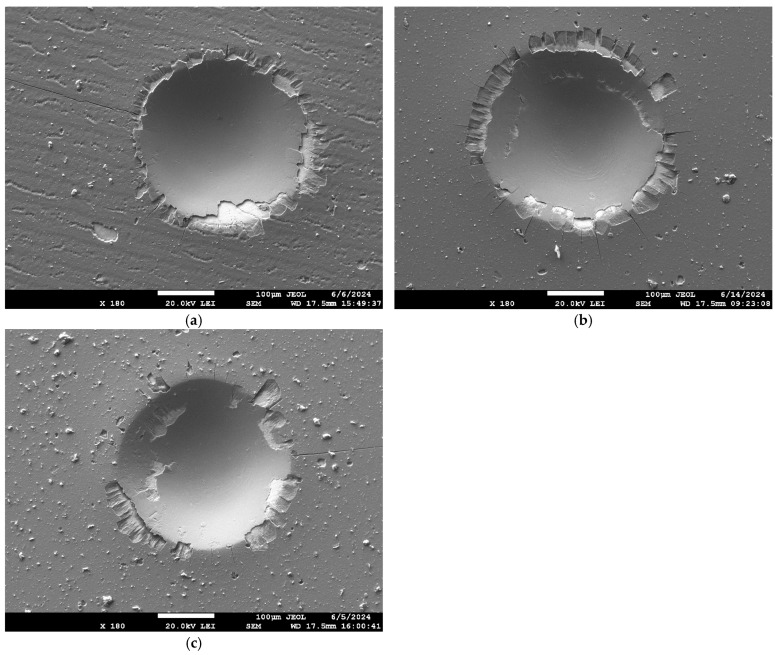
Examples of imprint images after the 1500 N load testing. (**a**) Sample 1A 1500 N 2nd imprint. (**b**) Sample 2A 1500 N 2nd imprint. (**c**) Sample 3A 1500 N 2nd imprint.

**Figure 4 materials-18-03777-f004:**
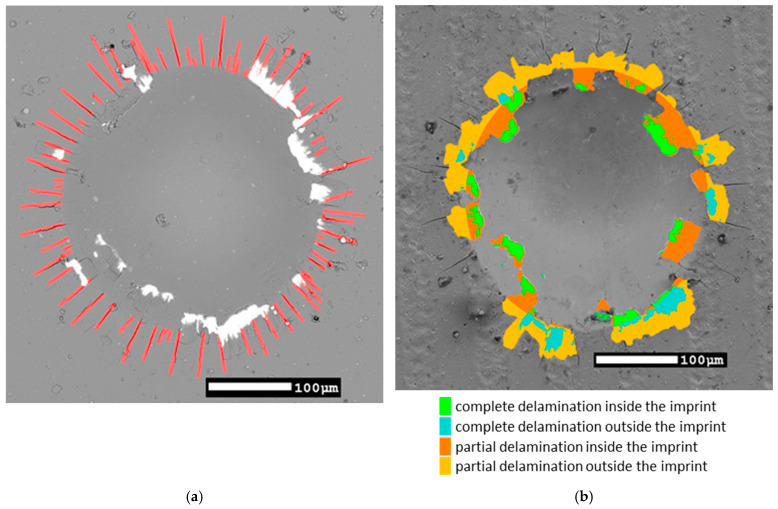
Representative SEM images of indents with key features marked: (**a**) Sample 3B 1500 N 2nd imprint—radial cracks; (**b**) Sample 1C 1500 N 2nd imprint—image masks of different delamination types.

**Figure 5 materials-18-03777-f005:**
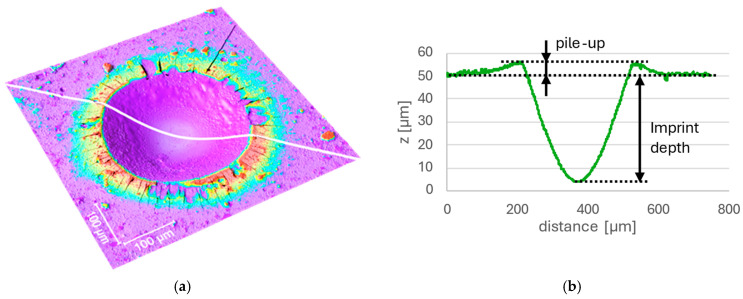
Representative LSCM data of Sample 3A 1500 N 2nd imprint: (**a**) Sample 3A 1500 N 2nd imprint—3D recreation of the imprint; (**b**) Sample 3A 1500 N 2nd imprint—height profile indicated by the white profile line in (**a**).

**Figure 6 materials-18-03777-f006:**
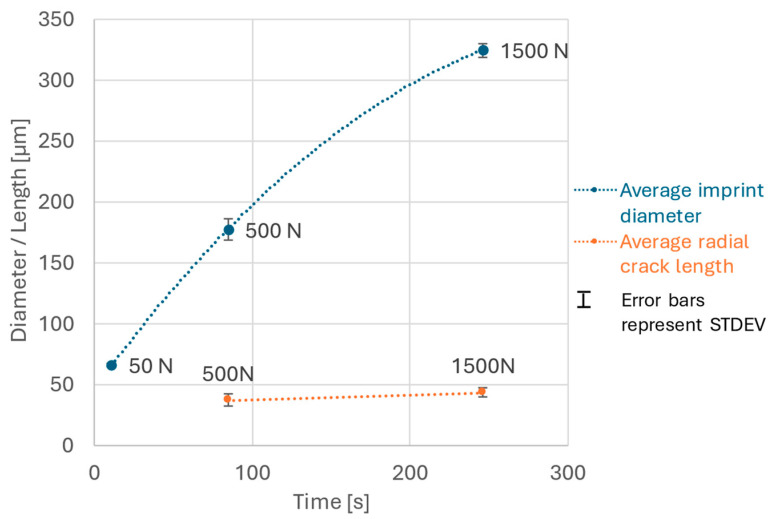
Average size of imprint features in relation to the duration of the test.

**Figure 7 materials-18-03777-f007:**
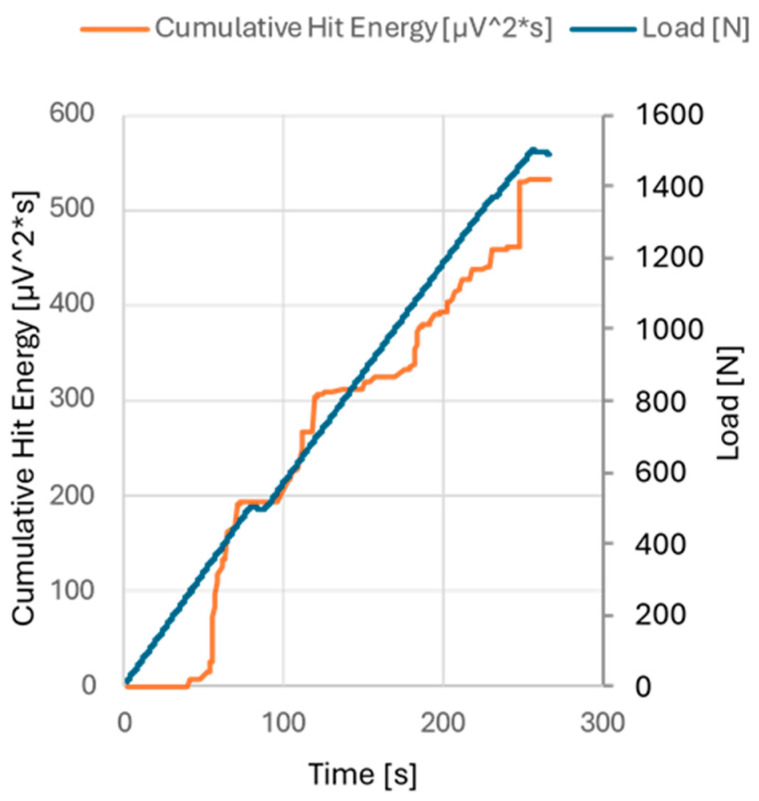
Cumulative hit energy chart for the sample 2A, 1st imprint.

**Figure 8 materials-18-03777-f008:**
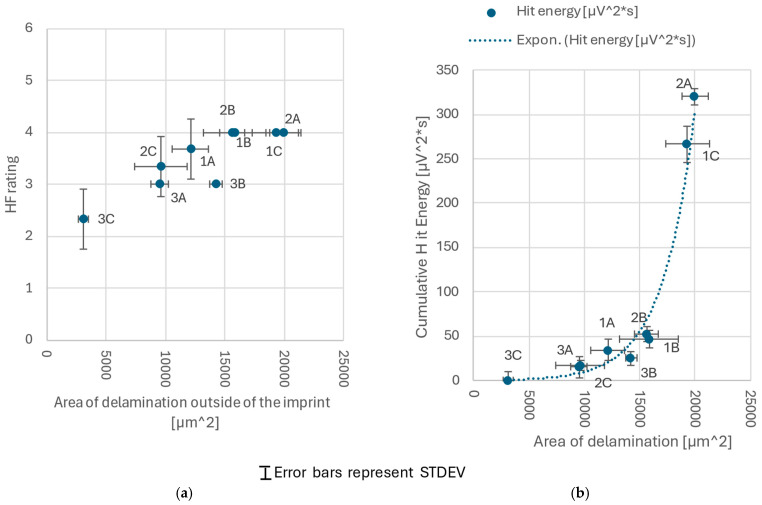
Correlations of delamination area outside of an imprint vs. (**a**) HF classification and (**b**) AE cumulative hit energy.

**Table 1 materials-18-03777-t001:** Laser ablation parameters.

	Laser Spot Speed v [mm/s]			Average Power P [W]	Repetition Rate f [Hz]	Pulse		
	A	B	C			Energy E [mJ]	Duration t [ns]	Peak Power P [kW]
1	1500	2000	2500	20	30,000	0.665	240	14
2	1500	2000	2500	20	76,000	0.265	50	9
3	1500	2000	2500	12	76,000	0.159	50	5.4

**Table 2 materials-18-03777-t002:** CrN PVD coating parameters.

Deposition Parameters	
Cathode current I_Cr_	150 A
U_s_	−120 V
Substrate temperature	450 °C
Working pressure	4 Pa

## Data Availability

The original contributions presented in this study are included in the article. Further inquiries can be directed to the corresponding author.

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
