# Peer review of "Characterization of Hard Coatings Using Acoustic Emission"

_materials, 2025, doi:10.3390/ma18163777_

Round 1
Reviewer 1 Report
Comments and Suggestions for Authors
All the comments and questions that I raised after the first submssion of this paper have been properly addressed by the authors. I would like to thank the authors for carefully revising everything. In my opinion the paper could now be accepted and published.
Reviewer 2 Report
Comments and Suggestions for Authors
In this paper authors talk about using AE technique to evaluate adhesion performance of the coating.
- In figure 2- authors can consider labeling x and y axis for better understanding of the graph
- Evaluation of Adhesion Properties of Hard Coatings by Means of Indentation and Acoustic Emission (https://doi.org/10.3390/coatings11080919) is another paper published in MDPI coatings journal. How does this particular research paper is different from MDPI coatings paper published in 2021.
- Authors can consider adding "Evaluation of Adhesion Properties of Hard Coatings by Means of Indentation and Acoustic Emission" as reference
Reviewer 3 Report
Comments and Suggestions for Authors
- Underscore the scientific value-added to your paper in your abstract.
Your abstract should clearly state the essence of the problem you are addressing, what
you did and what you found and recommend.
- In the section of Introduction, the authors need to mention the specific
research objectives and justify the novelty of this work, and reveal how their work is
different from prior reports?
In the introduction, the reference “Surface & Coatings Technology, 2015, 281: 176-183” should be cited for “Hence the results could be easily compared with available literature data. Multiple factors, such as mechanical interlocking, influence the adhesion between the coating and substrate. Controlling the surface roughness can significantly influence coating adhesion. In many cases, increased substrate roughness has been reported to enhance adhesion [16-19].”. In this reference, the adhesion of Ni-P coating also was improved via mechanical interlocking.
- How many measurements were carried out for substrate and final coating surface roughness values? How many samples were generated for each parameter? This must be mentioned in the experimental section, since authors have presented only a single value for these Average values for surface roughness must be presented along with standard deviation.
- The morphology in Fig. 3 possesses a certain degree of randomness. It can be influenced by selected regions and surrounding environment.
- A comparison of present data with data on similar work from literature must be presented.
- Please make sure your conclusions' section highlights the findings of the experiment. Highlight the novelty of your study.
Reviewer 4 Report
Comments and Suggestions for Authors
- The novelty of the work has not been specified. More recent relative works should be cited.
2.There is no indication of the number of repetitions conducted, and error bars are not provided in the figures.
- The authors said there was an exponential relationship between the AE signal energy and delaminated coating area. Why they present this correlation? The authors should give a more scientific explanation.
- In Figure 8b, a more detailed mathematic equation is encouraged to better present the relationship between the cumulative hit energy and area of delamination outside of an imprint, and the physical meaning of all the parameters in this equation should be supplied.
- Please consider citing some papers that published in the journal of “Materials”.
The English could be improved to more clearly express the research.
Round 2
Reviewer 3 Report
Comments and Suggestions for Authors
The manuscript has been carefully revised and can be accepted.
Reviewer 4 Report
Comments and Suggestions for Authors
the required problems have been well addressed